



# Changes in drought features at European level over the last 120 years

## Monica Ionita[1*] and Viorica Nagavciuc[1,2]

[1] Alfred Wegner Institute Helmholtz Center for Polar and Marine Research, Bremerhaven, Germany

[2] Forest Biometrics Laboratory – Faculty of Forestry, "Stefan cel Mare" University of Suceava, Suceava, Romania

[*] Corresponding author: Monica Ionita (Monica.Ionita@awi.de)

## Abstract

10    In this study we analyze the drought features at European level over the period 1901 – 2019, using three drought indices: the Standardized precipitation (SPI), the Standardized Precipitation Evapotranspiration Index (SPEI) and the self-calibrated Palmer Drought Severity Index (scPDSI). The results based on the SPEI and scPDSI indices point out to the fact that central Europe (CEU) and the Mediterranean region (MED) are becoming dryer, due to an increase in the potential evapotranspiration and the mean air
15    temperature, while the northern part of Europe (NEU) is becoming wetter. By contrast, the SPI drought index does not reveal these changes in the drought variability, mainly due to the fact that the precipitation does not exhibit a significant change, especially over CEU. SPEI12 indicates a significant increase both in the frequency and area over the last three decades for MED and CEU, while SPI12 is not capturing these features. By analyzing the joint probability of compound events (e.g. high temperatures/droughts),
20    we show that the potential evapotranspiration and the mean air temperature are becoming essential components for drought occurrence over CEU and MED. This, together with the projected increase in the potential evapotranspiration under a warming climate, has significant implications concerning the future occurrence of drought events, especially for MED and CEU regions.



## 1. Introduction

Over the last two decades, droughts have affected more then 2 billion people globally, and their impacts are increasing (CRED and UNISDR, 2019; IPCC, 2014, 2018; Van Lanen et al., 2016). Since the beginning of the 21$^{st}$ century Europe has become a "hot spot" for high intensity droughts and most of the European countries have suffered significant socio-economic losses (CRED and UNISDR, 2019; Spinoni et al., 2016b; Stahl et al., 2016). Drought events imply a series of risks to the environment and socio-human activities, and the way they are managed directly influences the final drought's costs. Overall, the impacts of drought events are felt over different sectors ranging from society, economy, forestry, biodiversity and agriculture. For example, the record breaking heatwave and the drought event in 2003, over Europe, has put an enormous stress on society, economy, environment and bio-diversity (Beniston and Stephenson, 2004). Vegetation growth across Europe was reduced by ~30% (Ciais et al., 2005), while crops and forests were much less productive than normal. Overall, summer 2003 heatwave and drought had a direct economic impact of ~17.134 billion Euro (CRED and UNISDR, 2019). Summer 2015 was the warmest and driest summer since 1950 over central and eastern part of Europe and the economic impacts of this event were estimated at ~2.172 billion (Ionita et al., 2017; Van Lanen et al., 2016). The economical and societal damages in 2015 were much smaller compared to the ones in 2003 mainly due to a better management at country and European level. Over the period 2018-19 more than 50% of central Europe was affected by drought, with significant consequences for economy, society and biodiversity (Bakke et al., 2020; Hari et al., 2020; Ionita et al., 2020; Ionita and Nagavciuc, 2020; Schuldt et al., 2020). The 2018 drought event extended also over the Nordic countries, leading to intense and devastating wild fires, with Sweden recording a record breaking burnt area of ~24310 ha (Bakke et al., 2020).

In simple terms, drought is seen as a direct consequence of precipitation deficit (Palmer, 1965), however drought intensity varies both in time (McKee et al., 1993) and in space (Vicente-Serrano et al., 2010). However, the World meteorological organization (WMO) has classified drought in four distinct categories: i) meteorological drought – defined as a rainfall deficit relative to a climatological norm; ii) agricultural drought – which is defined relative to the soil water availability during the growing seasons; iii) hydrological drought – which is associated with low water and ground water levels and low water supply to the reservoirs; the hydrological drought follows usually after many months of meteorological drought; and iv) socioeconomic drought – occurs when the water shortages start to affect people and takes into account the impact of drought conditions. Since drought is a very complex phenomenon, it is challenging to objectively quantify drought variability (Dai, 2011; Spinoni et al., 2016a; Vicente-Serrano



et al., 2010). Among the available drought indices, the Palmer Drought Severity Index (PDSI), the
Standardized Precipitation Index (SPI) and the Standardized Precipitation Evapotranspiration Index
(SPEI) are most commonly used. While PDSI (Palmer, 1965) was successful in quantifying drought

variability and severity in the United States, it was less successful for other regions (e.g. Australia and
South Africa) (Burke et al., 2006; Ntale and Gan, 2003). In this respect, the self-calibrated palmer
Drought Severity Index has been developed (Wells et al., 2004), which calibrates automatically the
behavior of the index for each location, by replacing empirical constants in the index computation with
dynamically calculated values. Nevertheless, one of the major drawbacks of PDSI/scPDSI is that it does

not have a multi-scalar character (e.g. it cannot be computed for different time scales). To overcome the
multi-scalar feature, the SPI has been developed (McKee et al., 1993), which relies on a probabilistic
precipitation approach. The multi-scalar character  allows us to analyze the effect of precipitation deficit
on the different water-resources components on different time scales. SPI takes into account just the
precipitation variability, while the role of temperature is ignored, which under the current and projected

climate change, can be a limiting factor for drought risk management. To take into account also the role
of temperature, Vicente-Serrano et al. (2010) have developed the SPEI. The main advantage of SPEI is
that incorporated the multi-scalar character of SPI with the ability of including the effect of temperature
on drought assessment. Tacking into account that none of the aforementioned drought indices are
inherently superior to the rest in all circumstances, one might perform better than the others in terms of

providing useful information for drought monitoring and forecasting over different regions (Stagge et
al., 2017; Wang et al., 2020). Thus, comparing and analyzing the three drought indices (scPDSI, SPI and
SPEI) together can be helpful in understanding which one has the ability to monitor drought feature and
evolution over different regions, and since different drought indices used different input parameters,
complex analyses of all drought indices will allow determining the main parameters which affect drought

occurrence. In previous studies it has been shown that SPI identifies the drought 1 month earlier than
PDSI (Hayes et al., 1999), and the correlation between PDSI and SPI and SPEI is higher in semiarid
regions than in humid ones. Usually the highest correlation is obtained between PDSI and SPI and SPEI
for longer time scales (9 – and 12-monthly time scales) (Paulo et al., 2012; Vicente-Serrano et al., 2010),
thus, in this study we employ SPEI and SPI for an accumulation period of 12-months.

Climate projections indicate that Europe will be one of the future hot spots for hydro-climatic change
with the southern and central part of Europe getting drier and the northern part getting wetter (Cook et
al., 2020; IPCC, 2018; Naumann et al., 2018; Spinoni et al., 2018, 2020). Therefore, a better
understanding of drought characteristics at European level and at macro regions (e.g., the Mediterranean





region, Central Europe and the northern Europe) is crucial for a better drought monitoring and

forecasting, in order to provide reliable adaptation strategies for drought hazard. The drought events over the last two decades were not homogenously distributed throughout Europe, and each event had a specific center of action, and the drought centers of action have move/migrated. Thus, a detailed analysis of the drought evolution at a regional level, over the last century, is needed. In this respect, here we analyzed the variability of droughts, over the last 120 years, over three key macro regions, as defined by the IPCC:

the South Europe/Mediterranean region (MED), Central Europe (CEU) and North Europe (NEU). In the current study we want to extend on previous studies (Spinoni et al., 2015, 2017) and make an updated and in-depth analysis of the drought characteristics, at European level, for the last ~120 years. Compared to previous studies (Spinoni et al., 2015, 2017; Vicente-Serrano et al., 2021), here we make a direct comparison between three different drought indices (SPEI, SPI, and scPDSI), each with its specific

advantages/disadvantages and we extend the analysis until the end of 2019. This is a very important aspect of our study, tacking into account that the drought event 2018-19 set a new European drought benchmark (Hari et al., 2020). This paper is structured in 4 main sections, including the introduction. In Section 2 the data and methods used in this study are presented, while in section 3 we make a detailed description of the results of our study. In Section 4 the main conclusions and outcomes of the paper are

presented.

## 2. Data and methods

As stated before, the main region of analysis for this study is Europe, but for most of the analyses employed through the paper we have splitted the European domain in three separate macro regions (Iturbide et al., 2020). These regions, which were chosen following the recommendation from the 5[th]

Assessment Reports of the IPCC (IPCC, 2014) are: a) the South Europe /Mediterranean region (MED); b) Central Europe (CEU) and c) North Europe (NEU) (*Figure S1*).

The monthly precipitation amount (PP), monthly mean air temperature (TT), and the Potential Evapotranspiration (PET) used in this study are obtained from the CRU TS v. 4.04 version dataset (Harris et al., 2020). All analyzed data cover the 1902 – 2019 period and have a spatial resolution of 0.5° x 0.5°.

For the drought analysis we have used three drought indices: the Standardized Precipitation Index (SPI), the Standardized Precipitation Evapotranspiration Index (SPEI) and the self-calibrated Palmer Drought Severity Index (scPDSI). All indices are computed based on the PP, TT and PET data from the aforementioned CRU TS v. 4.04. SPI takes into account the accumulated precipitation data, where the PP data has been fitted to a gamma distribution (McKee et al., 1993). The SPEI index computation is





based on the probability distribution of the difference between PP and PET (PP - PET). The data is normalized into a log-logistic probability distribution to obtain the SPEI index (Vicente-Serrano et al., 2010). The potential evapotranspiration data was computed by employing the Penman – Monteith equation (Vanderlinden et al., 2008). One of the most important advantages of the SPI/SPEI is the representation of multiple time scales, which allows the monitoring of different drought types, such as:

meteorological, agricultural and hydrological. Having a multiscalar characteristic, both SPI and SPEI have been computed for different time scales (e.g. 1, 3, 6, 9 and 12 months). Negative values of SPI and SPEI indicated dry conditions, while positive values indicate wet conditions. For the current study we have use three different classes of drought (Lloyd-Hughes and Saunders, 2002): i) moderate drought (SPI/SPEI values between -1 and -1.5); ii) severe drought (SPI/SPEI values between -1.5 and -2), and

iii) extreme drought (SPI/SPEI values less than -2). Both SPI and SPEI have been calculated using the R-package SPEI (https://cran.r-project.org/web/packages/SPEI/index.html).

The scPDSI index is based on the well-known Palmer Drought Severity Index (PDSI). Nevertheless, because of data limitations and regionalization used to derive the weighting and calibration algorithm, the original PDSI is not suitable for all regions (Burke et al., 2006). In this respect, here we use the

scPDSI index which automatically calibrates the behavior of the index at different locations by replacing the empirical constant with dynamically calculated values (Wells et al., 2004). As in the case of SPI and SPEI, we have defined also for scPDSI three different drought classes: i) moderate drought (scPDSI values between -2 and -3); ii) severe drought (scPDSI values between -3 and -4) and iii) extreme drought (scPDSI values less than -4).

To test the influence of TT and PET on the probability of occurrence of dry events, we employ a joint distribution analysis of compound events (e.g. the co-occurrence of low precipitation and dry events or high temperature and dry events) (Hao et al., 2019). In this study we focus on the SPEI for an accumulation period of 12-months (SPEI12), PET, PP and TT averaged over the three regions: MED, CEU and NEU. For each region and each two variables (e.g. PP and SPEI12, PET and SPEI12 and TT

and SPEI12) we computed a binary variable (Y =1 for co-occurrence and Y =0 for non-occurrence) which indicates the occurrence based on PP/PET/TT and SPEI12. For specific threshold of the variables, the occurrence of a compound events can be expressed as:

$$Y = \begin{cases} 1, & P \le px, T > tx \\ 0, & otherwise \end{cases}$$



Where *px* indicates the precipitation threshold and *tx* indicates the temperature threshold, for example.
For the current analysis we have chosen as threshold the 80th percentile for TT and PET and the 20th
percentile for SPEI12 and PP.

**3.1 Drought trends over the last 120 years**

The spatial patterns of the Mann-Kendall trend statistics (Mann, 1945) are presented in Figure 1 for the
December SPEI12 (*Figure 1a*), December SPI12 (*Figure 1b*) and the annual scPDSI (*Figure 1c*) for the
1902 – 2019 period. Positive values indicate a trend towards wetter conditions, while negative values
indicate a trend towards drier conditions. SPEI12 exhibits a very clear signal: most of the countries from
MED and CEU show a significant decreasing trend (drying) over the last 120 years, while the countries
from NEU exhibit a significant positive trend (wetting) (*Figure 1a*). SPI12 exhibits significant and
negative (drying) trends only over small regions over CEU (e.g. Czech Republic, Slovakia, Hungary,
Belarus, and Poland) and over MED (Italy, southern Spain, Albania and Greece) and a positive trend
(wetting) over NEU (*Figure 1b*). Similar results, based on SPI12, have been found by Vicente-Serrano
et al. (2020). In their study extending back to 1851, Vicente-Serrano et al. (2020) have shown that SPI12
exhibits positive trends over U.K. and central Europe, and negative trends over Italy and the Balkans.
The results based on the annual scPDSI index are similar as the ones observed for SPEI12: a significant
drying trend for MED and CEU, with small exceptions over Ukraine and Turkey and a significant wetting
trend over NEU (*Figure 1c*).

At shorter time scales (e.g. 3 months) there is a clear seasonal signal in the evolution of the drought
phenomenon. During winter (February SPEI3 and SPI3), NEU and large parts of CEU, except the Czech
Republic, are characterized by a wetting trend over the last 120 years (*Figure S2a and S2b*), while for
MED no significant trend is observed. In spring, May SPEI3 indicates a significant drying trend over
most of the countries in the MED region and over the eastern part of CEU region and a wetting trend in
the northern part of NEU (*Figure S2c*). May SPI3 shows a different perspective: no significant (wetting
or drying) trend is observed in CEU and MED. For NEU, May SPI3 captures the same features like May
SPEI3: a significant wetting trend over the north part of NEU (*Figure S2d*). August SPEI3 features a
significant drying trend over MED and CEU, with the highest drying amplitude over the Iberian
Peninsula, and a significant wetting trend over the northern part of NEU (*Figure S2e*). The significant
drying trend over MED and CEU are not visible in August SPI3, but the wetting trend over the northern
NEU, is captured by August SPI3, similar with August SPEI3 (*Figure S2f*). In autumn, both November





SPEI3 and SPI3 indicate a significant wetting trend over NEU, and no significant changes over MED
and CEU (*Figure S2g and S2h*).

From the analyses above, we can see that there are differences in the drought evolution over the last 120
years as reflected by the SPI and SPEI /scPDSI, especially over MED and CEU. This might be due to
the fact that in the computation of the SPEI index the potential evapotranspiration, hence temperature, is

included. To test the influence of PET and TT variability on the difference observed between SPEI and
SPI, we have computed also the seasonal PET, TT and PP trends over the European region (*Figure S3*).
PET is characterized by a significant positive trend (increased potential evaporation) over MED and CEU
and the southern part of NEU in spring (*Figure S3d*) and summer (*Figure S3g*), with the highest
amplitude in summer over the MED and CEU. A positive and significant trend is observed also in autumn

(*Figure S3j*), but just over the western part of CEU and over MED. The seasonal precipitation trends
follow the same pattern as those obtained of the seasonal SPI3 index: a significant wetting trend over
NEU in all seasons (*Figure S3b, S3e, S3h and S3k*). In spring, summer and autumn no significant
precipitation changes are observed over MED and CEU. In the case of the seasonal mean air temperature,
the trend signal is very clear: in all seasons there is a significant warming over all analyzed regions

(*Figure S3c, S3f, S3i and S3l*). In winter and spring, the warming with the highest amplitude is observed
over the eastern part of Europe, while in summer the highest amplitude is observed over the Iberian
Peninsula and Austria.

**3.2 Drought area**

Europe has experienced a number of extremely dry summers within the last decade (e.g. 2015, 2018,
2019) which have been already documented in previous studies (Bakke et al., 2020; Hari et al., 2020;
Ionita et al., 2017; Laaha et al., 2017). To put the last decade drought events into a longer perspective,
we have computed the drought area for MED, CEU and NEU, affected by three types of drought:
moderate (SPEI12/ SPI12 between -1 and -1.5, and scPDSI between -2 and -3), severe (SPEI12/ SPI12

between -1.5 and -2, and scPDSI between -3 and -4) and extreme (SPEI12/ SPI12 smaller than -2, and
scPDSI smaller than -4), considering the 12-month SPEI (December SPEI12) and SPI (December SPI12)
indices and the annual scPDSI index. For MED region a significant increase in the area affected by all
types of drought can be observed for SPEI12, SPI12 and scPDSI (*Figure 2, Table S1*). The years with
the largest area affected by all types of drought (based on SPEI12 and scPDSI) were recorded over the

last decade, the peak being observed over the period 2016-17 (*Figure 2a and 2c*). The year with the
largest affected area by drought, based on SPI12, was 1946-47 (*Figure 2b*). Overall, the amplitude of the



drought area is underestimated by SPI12 compare to SPEI12 and scPDSI over the last ~30 years, since the SPI12 does not take into account the temperature variability.

In the case of CEU region, the driest years based on SPEI12 (*Figure 3a*) and scPDSI (*Figure 3c*), in terms of spatial coverage (~95% / 71% / 34%) affected by moderate/sever/extreme drought are: 1920–1921, 1976, 2015 and 2018-19, when was recorded the largest affected area. As in the case of MED, the drought events over the last ~three decades are underestimated when we take into consideration SPI12 (*Figure 3b*). The driest years based on SPI12, in terms of the largest spatial coverage (~95% / 78% / 45%) affected by moderate/sever/extreme drought are 1954 and 1976, with the maximum spatial coverage in 1920 – 1921. While for MED there was a significant increase in the area affected by drought over the last ~120 years, in the case of CEU there are altering periods of intense dryness and wetness, with a spatial coverage of almost ~90% characterized by prolonged drought conditions, and periods of no drought or reduced drought in term of spatial coverage. There are significant and positive trends, in the spatial extend of all types of droughts for SPEI12 and scPDSI, and significant and negative trends for SPI12 (*Table S1*).

The spatial coverage of droughts for NEU shows a relatively different picture compared to MED and CEU. Over the last 30 years there are relatively fewer drought events recorded and their spatial extent is rather small compared to the ones from the beginning of the 20[th] century (*Figure 4*). For the NEU region, SPEI12, SPI12 and scPDSI show a rather similar variability: higher spatial extent of drought events between 1900 – 1922, 1935 – 1950, 1959 – 1962 and 1970 – 1980. The driest years, in terms of spatial coverage, are: 1909, 1940 – 42, 1947, and 1976. The spatial coverage, for all types of drought, shows a significant and negative trend for all analyzed indices (SPEI12, SPI12 and scPDSI, *Table S1*).

### 3.3 Drought duration maps

To provide a complete picture of the drought hot spots, over the last ~120 years, we have splitted the data set in twelve different time periods, covering each decade since the beginning of the 20[th] century up to the end of 2019. We choose these periods to have an equal number of months/years (120 months/10 years) for all the analyzed periods. The only exception are the beginning and the end of the data set: the 1902 – 1910 and 2011 – 2019 periods where we have 108 months and 9 years for each mentioned decade.

The aim of splitting the data in short time periods was to test if there were significant changes in the drought conditions on decadal time scale. The analysis is performed for SPEI12 and SPI12 for three different drought categories, as in the previous section: moderate (SPEI12/ SPI12 between -1 and -1.5), severe (SPEI12/ SPI12 between -1.5 and -2) and extreme (SPEI12/ SPI12 smaller than -2). The drought



frequency in each category (moderate— *Figure 5/ S4*, severe—*Figure 6/ S5* and extreme—*Figure 7/ S6*)

is expressed as the number of months/time period in a given category when SPEI12 and SPI12 were below a certain threshold.

In terms of moderate drought, based on SPEI12, the decades characterized by a high frequency of dry events (more than 40 months/10 years) are: 1941 – 1950, 1971 – 1980, 2002 – 2010 and 2011 – 2019 (*Figure 5*). Over the 1941 – 1950 decade, the drought hot spots are over the central, eastern and northern

parts of Europe, the only exception being the countries around the eastern part of the Baltic Sea (e.g., Poland, Lithuania, Latvia and Estonia). Over the 2011 – 2019 decade, the drought hot spot is localized over MED and CEU.  Over the first eight decades of our analyzed period (1902 – 1980), the northern part of Europe was characterized by a relatively high frequency of dry events, when compared with the last four decades of our analyzed period, for which the frequency of dry events is very low. Over the last

120 years, the European regions was characterized by the different spatial distribution of the moderate drought hot spots  based on the SPI12 maps (*Figure S4*). The driest decades, based on SPI12, are: 1902 – 1910, 1941 – 1950, 1971 – 1980 and 1981 – 1990. Over the last three decades of the analyzed period, there is a clear reduction in the frequency of dry events over almost all analyzed regions (*Figure S4*). The driest decade is 1941 – 1950, when most of the European regions recorded up to 60 months/10 years

of moderate drought.

In terms of severe drought, based on SPEI12, the decades characterized by a high frequency of dry events (more than 25 months/10 years) are: 1941 – 1950, and 2011 – 2019 (*Figure 6*). Over the 1941 – 1950 decade the drought hot spots are over the central Europe (e.g., northern Italy and southern part of Germany, Croatia, Romania and Ukraine), the southern part of Norway and Finland. Over the 2011 –

2019 decade, the severe drought hot spot is localized, as in the case of the moderate drought, over MED and CEU. The driest decades in term of drought duration according to SPI12 maps are 1911 – 1920 over the northern part of Fennoscandia, 1941 – 1950 over MED, CEU and NEU, except the countries around the eastern side of the Baltic Sea, and 1981 – 1990 over a region stretching the eastern part of Europe (*Figure S5*). Overall, throughout the analyzed period, there is an inhomogeneous evolution of the severe

drought hot spots.

*In Figure 7*, the hot spots representing the extreme drought events are shown. For each decade covering the period 1902 – 2000 there relatively just few months (up to 10months/10years) when extreme drought conditions were recorded, over different small regions throughout the European continent. Over the 2001 – 2010 decade a hot spot of extreme drought can be observed, based on SPEI12, mostly over the eastern

part of Europe. The 2011 – 2019 decade is characterized by a high frequency of extreme dry event over



MED and CEU, the hot spots being over Germany, Czech Republic, Spain and Italy. The frequency distribution of the extreme drought based on the SPI12, shows different results. The frequency of extreme dry events over the last three decades is very small or non-existent over all analyzed regions (*Figure S6*). Opposite to this, there is a higher frequency of dry events over central Europe and the eastern most part

of Europe over the 1921 – 1930 decade and a relatively high frequency of dry events over Sweden and the southern part of Europe.

### 3.4 Compound events: PP vs. TT vs. PET

As previously mentioned, due to the consideration potential evapotranspiration, hence of temperature, in

the computation of SPEI, the drought index reflected by the SPEI indicated a significant drying trend over MED and CEU at various timescales (e.g., 3-months and 12-months), while the drought index reflected by the SPI showed opposite or no changes over these two regions. Moreover, we have found a significant increase both in the frequency and spatial extent of dry events over the last two decades over MED and CEU, when using the SPEI12, and opposite results when using SPI12 index, which is solely

based on the precipitation variability. To emphasize the influence of PP, PET and TT on the variability of the SPEI12, in *Figures 6-8* we have computed the changes in occurrence of concurrent extremes (e.g., the low precipitation/drought, high temperature/drought, high evapotranspiration/drought), by averaging the annual PP, TT, PET and SPEI12 over each region (MED, CEU and NEU).

In the case of MED region, the drought events which occurred before the 1990's have been driven mainly

by a precipitation deficit (*Figure 8a – green dots*). Starting with the 1990's the occurrence of dry events was influenced not only by PP, but also by changes in TT (*Figure 8a – red dots*) and PET (*Figure 8a – yellow dots*). For the years 1999, 2006, 2008, 2012, 2013, 2016 and 2019 the drought events have only occurred along with significant anomalies in TT and PET (*Figure S7*). For CEU, the co-occurrence between low precipitation and SPEI12 has been a permanent feature over the period 1902 – 1976. After

this period, the role of TT (*Figure 8b – red dots*) and PET (*Figure 8b – yellow dots*) becomes more important compared to the one of PP. For the years 1983, 1992, 2014, 2018 and 2019 the drought events, over the CEU region, have only occurred along with significant anomalies in TT (*Figure 8b – red dots*) and PET (*Figure 8b – yellow dots*). Over the last 40 years, there were just 3 years (e.g., 2003, 2011 and 2015) when the drought events were also accompanied by a precipitation deficit (*Figure S8*). In the case

of NEU, all dry years were accompanied by low precipitation years, except for the year 2018 (*Figure 8c – green dots*). The role of TT (*Figure 8c – red dots*) and PET (*Figure 8b – yellow dots*) in driving the occurrence of dry events was recorded just for 2 years: 1976 and 2018 (*Figure S9*). Thus, in the case of


NEU, the role of precipitation dominates the occurrence of dry years throughout the analyzed period. Overall, for MED and CEU there is a significant increase (99% significance level) in the probability of

co-occurrence of compound events related to warm/dry events and high evaporation/drought over the last three decades, and no significant change in the probability of occurrence of compound events over NEU.

**3.5 Rank maps and extreme dry events**

To analyze the extremeness and the spatial extent of the top 5 drought events, over Europe, we use the ranking map methodology (Bakke et al., 2020; Ionita et al., 2017). In this respect, we compute the ranking maps of SPEI12 and SPI12, for each month (e.g., January – December) over the 1902 – 2019 period. The five driest years (the lowest SPEI12 recorded at each grid point) from January to December are shown in *Figure 9*. To most striking feature of the rank maps is the persistence of the extreme drought events in

1921 and 2018-19 from January to December. March 1921 ranks as the driest one on record over the eastern part of Ukraine and the western part of Russia. This event continues to rank as the driest one on record, over the same region, until July 1921. From August 1921 until January 1922 this event ranks as the driest one on record, shifting his center from western Russia to the north part of France and southern part of U.K (*Figure 9a, 9j, 9k and 9l*). The evolution of the monthly SPEI12 index from November 1920

until January 1922 (*Figure S10*) indicates that this event had its origin over Ukraine and western part of Russia in the first months of the 1921 year than it moved westward towards Europe, reaching the highest amplitude over France and southern part of UK from November 1921 to January 1922 (*Figure S10*). The 1921 year was also the driest one on record, in terms of low flow, in the Rhine and Weser catchment areas (Ionita and Nagavciuc, 2020). The drought event in 1921-22 was driven mainly by a precipitation

deficit over the central and eastern part of Europe (*Figure 10a and S8*), and to a lesser extent by TT and PET. The spatial extent of the 1921-22 event is much higher if we take into account SPI12 compared to SPEI12 (*Figure 11 – left column*). This pattern can also be observed based on SPI12 index monthly rank maps(*Figure S12*). The SPI12 rank maps follow the same pattern as SPEI12 for the event 1921-22. In the case of extreme drought, the area affected by drought in 1921-22, based on the SPI12 index (*Figure*

*11c*), is almost double compared to the area cover by drought based on the SPEI12 index (*Figure 11b*). The year 2018 is captured as the driest year over the central part of Europe from November 2018 until August 2019 (*Figure 9*). This event affected all Europe, except the north part of Fennoscandia, with the highest amplitude over the north-eastern part of Germany. On shorter time-scales (e.g. SPEI3) the event starting developing already in spring 2018 (Bakke et al., 2020). On longer time scales (e.g., SPEI12) the





development of this event started towards the end of 2018 and it was mainly driven by record high temperatures and enhanced evaporation over the European region throughout the summer 2018 (*Figure 10d, 10f and S8*). This event persisted until the end of the summer season of 2019 (*Figure S13*), with a special focus on the north-eastern part of Germany (Hari et al., 2020; Ionita et al., 2020). The spatial extent of the 2018-19 event is much higher according to  the SPEI12 index (*Figure 11f*) and scPDSI

index (*Figure 11h*) compared to SPI12 index (*Figure 11g*).

Other extremely dry years, as captured by the rank maps based on SPEI12, are: 1947, 1976, 2003 and 2015. The year 1947 was extremely dry over Norway and Finland, from September 1947 up to December 1947 (*Figure 9j, 9k and 9l*). Overall, the summer of 1947 was dry throughout Europe, but Norway was especially hard-hit. Weather records from Oslo in July and August showed there were only 2.2 mm of

rain for an entire month while the monthly average is ~ 102 mm (Hisdal et al., 2006). Summer 1976 (SEPI12 – June, July, August and September) ranks as the driest on record over different regions extending from the southern part of U.K, western part of Germany and southern part of Norway (*Figure 9f, 9g, 9h and 9i*). The summer of 1976 was considered to be one of the hottest summers in Europe, mainly due to a long-lasting atmospheric blocking pattern which has dominated most of Europe for all

of the summer months (Rodda and Marh, 2011). The drought events in 2003 and 2015 were restricted mostly to the summer months, and they were driven by record braking temperatures and an extreme soil-moisture deficit (Ionita et al. 2017 and the references therein). We have computed also the rank maps also for the scPDSI index (*not shown*) and overall, the driest years captures by the monthly evolution of scPDSI are similar with the ones recorded by SPEI12 (e.g., 1921-22, 1947,1976, 2003, 2015 and 2018-

370    19).

## 4. Conclusions

In this study we have shown the importance of making comparatives analyses, at large spatial scales (e.g., Europe) based on different drought-related indices. The novelty of this study is represented by the

fact that we make an in-depth analysis of drought frequency and extent for three different drought indices (e.g., SPEI12, SPI12 and scPDSI) covering the 1902 – 2019 period, and we show that that after 1990's there is a significant divergence between SPEI/scPDSI and SPI, driven mainly by an increase in the mean air temperature and evapotranspiration. Changes in several drought characteristics are investigated based on data for the past 120 years, including percentage of area affected by drought and drought frequency.

Our results indicate that droughts over Europe exhibit significant differences depending on the type of drought index used. Based on SPEI12 index we observe a well-defined decadal variation of drought





events during the past 120 years, with more frequent droughts occurring between 1941 and 1950 and after 2000s, and fewer drought events in the 1900s and 1990s. Based on changes in affected drought area, several regional differences are detected. When tacking into account the SPEI12 drought index, the

observed changes from our study are in line with the suggested changes by future projections as an effect of climate changes, namely: a significant drying trend over MED and CEU as a response to an increase in the temperature and evapotranspiration and not necessarily a rainfall deficit (IPCC, 2018; McCabe and Wolock, 2015; Spinoni et al., 2018). For NEU, all indices indicate a wetting trend over the analyzed periods. Similar results have been also shown by Stagge et al. (2017), namely a significant deviation in

the drought area measured by SPEI and SPI, but their study was limited to a shorter period of time (1958 – 2015). Overall, our results indicated that the rainfall deficit contribution to drought occurrence is significant over NEU, while TT and PET are becoming, along with PP, essential ingredients for drought occurrence in MED and CEU. The contribution of TT and PET to drought occurrence, has become significant, especially after the 1990's both for MEU and CEU (*Figure 8*). The lack of significant

changes, when taking into account SPI, has been recently detected also by Vicente-Serrano et al. (2021). According to their results, for the western part of Europe no long-term changes in the drought occurrence could be detected, by using precipitation records alone, which is line with our findings.

Overall, the main conclusions of our study can be summarized as follows:

- The trend analysis, based on SPEI12 and scPDSI, indicates that most of the countries from MED and
CEU regions show a significant decreasing trend (drying) over the last 120 years, while the countries from NEU exhibit a significant positive trend (wetting). When we take into account SPI12, no significant changes are observed, except some small regions (e.g., the southern part of Poland, Czech Republic, Italy and southern Spain). As expected, the trend observed for SPI12 (*Figure 1b, Figure S2 - right column*) follow the trends observed for the seasonal precipitation (*Figure S3 – middle*
*column*).

- The analysis based on the drought duration map, indicates that there is an increase in the frequency of moderate, severe and extreme droughts, based on SPEI12, over CEU and MED over the last two decades. The analysis based on the SPI12 index, indicates a rather opposite pattern: a reduction in the frequency of dry event over the last two decades, especially in the case of extreme droughts, over
most of the European region.

- Based on the joint distribution of compound events (e.g., the co-occurrence of low precipitation and dry events or high temperature and dry events) we show that CEU and MED have changed from a rainfall deficit dominated drought risk to a more temperature dominated drought risk, especially over





the last two decades, and PET and TT are becoming essential ingredients for drought occurrence over
MED and CEU.

- The drought events of 1920/21 and 2018/19 are the most extremes one in terms of spatial extent and amplitude (*Figure2, Figure 3 and Figure 9*) over the last 120 years. While the 1920/21 event was driven mainly by a significant rainfall deficit, the 2018/19 event (the second most extreme) was driven mainly by extremely high temperatures and increased evaporation rates.

- Due to the consideration potential evapotranspiration, hence of temperature, in the computation of SPEI, the drought reflected by the SPEI showed a drying trend over MED and CEU at various timescales, while the drought reflected by the SPI shows opposite or no changes. Thus, the performance of the SPI may be insufficient for drought analysis studies over regions where there is a strong warming signal.

Therefore, in this study we highlight the importance of temperature, hence of the potential evapotranspiration in delineating the drought spatio-temporal variability and we provide a vital reference for the applicability, at European scale, of the SPEI, SPI and scPDSI under climate change. SPEI and scPDSI indicate an increase in drought area and occurrence frequency, for MED and CEU, which are mainly induced by a significant increase in TT and PET. By contrast, the SPI does not reveal these
features, for MED and CEU since the precipitation does not exhibit a significant change. The only region where all indices indicate the same changes, namely a wetting trend, is NEU. Based on the results obtained from this study, we suggest that the increasing mean air temperature and the potential evapotranspiration can amplify drought, over the southern and central part of Europe, thus it has implications concerning the future occurrence of drought events, given that potential evapotranspiration
is project to increases under a warming climate. In this respect, the spatial extent and the duration of the 2018/19 event can be an indication that the climate changes signal is already producing palpable effects in the south and central part of Europe, in concordance with the project climate change signals for Europe (Naumann et al., 2018; Spinoni et al., 2018). Therefore, the SPEI is probably a more suitable index than the SPI to study the spatio-temporal variability of drought in Europe under climate change, especially for
MED and CEU regions.






**Acknowledgements.** MI was supported by Helmholtz Association through the joint program "Changing Earth - Sustaining our Future" (PoF IV) program of the AWI. Funding by the AWI Strategy Fund Project - PalEX and by the Helmholtz Climate Initiative - REKLIM is gratefully acknowledged. VN was supported partially by the project number PN-III-P1-1.1-PD-2019-0469.


**Author contribution.** MI designed the study and wrote the paper. VN analyzed part of the climate data and helped write the paper and interpret the results.

**Data availability.** The data that support the findings of this study are available from the corresponding author upon reasonable request.


**Competing interests.** The authors declare that they have no conflict of interest.











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

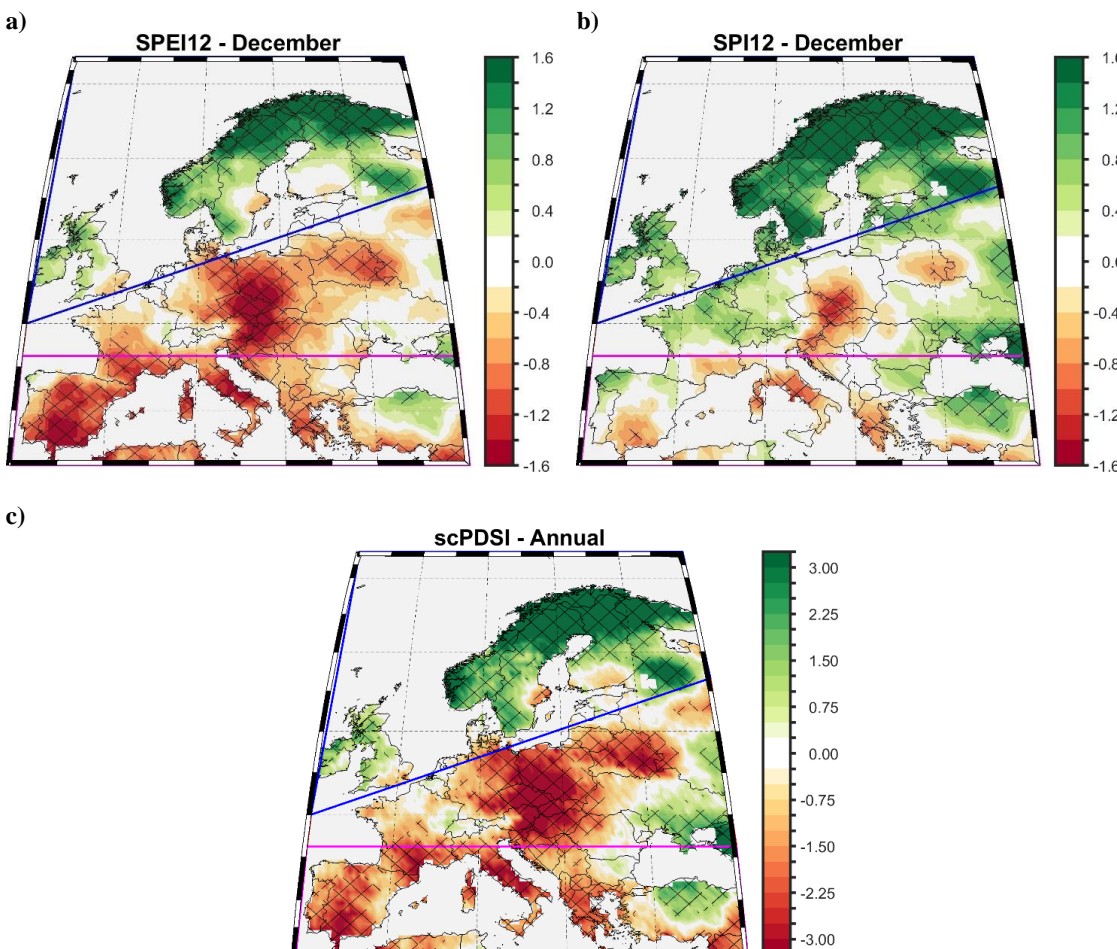

*Figure 1*. a) Linear trend of the December SPEI12; b) as in a) but for SPI12 and c) as in a) but for the annual scPDSI. Stipples indicate statistically significant trends. Period 1902 – 2019. Units: z-scores/ 118 years.



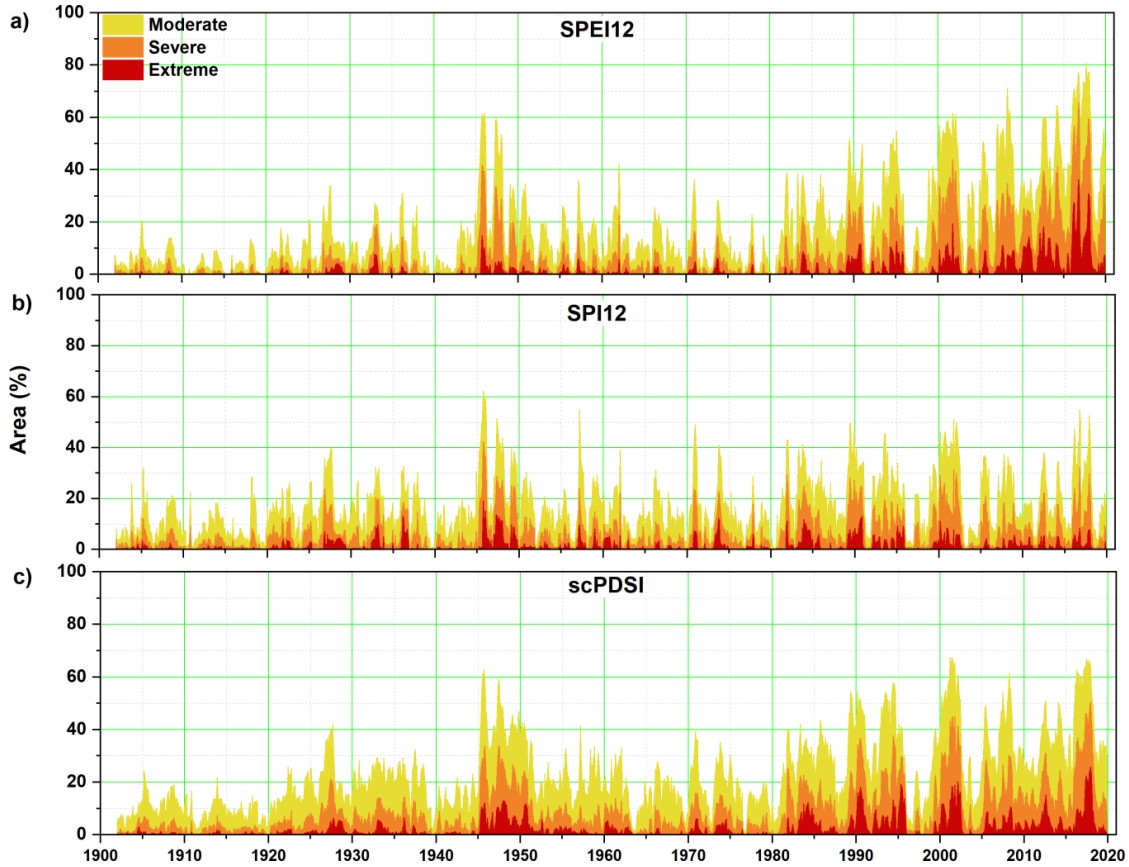

**Figure 2.** Temporal evolution of the percentage area affected by droughts over **MED** for: a) SPEI12, b) SPI12 and c) scPDSI, for three drought severity categories: moderate (yellow), severe (orange) and extreme (dark red). See text for the definition of the drought categories.



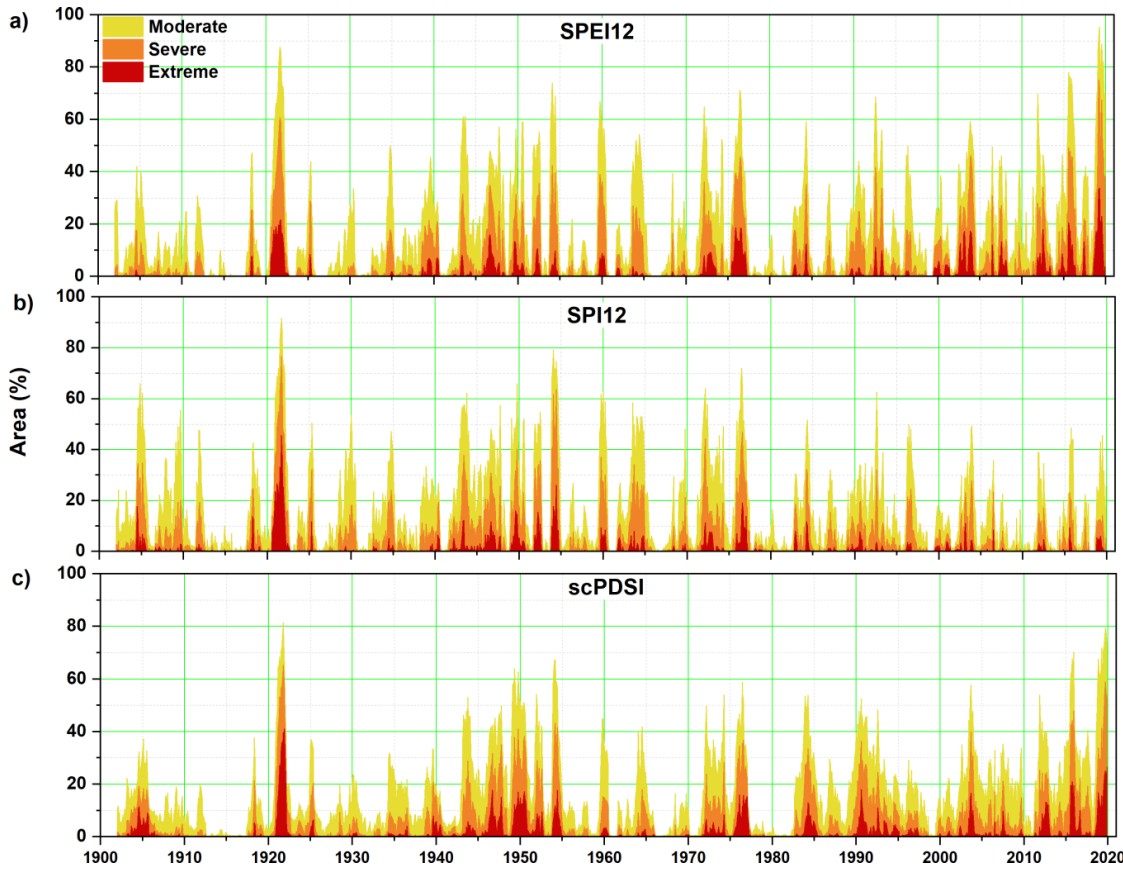

**Figure 3.** Temporal evolution of the percentage area affected by droughts over **CEU** for: a) SPEI12, b) SPI12 and c) scPDSI, for three drought severity categories: moderate (yellow), severe (orange) and extreme (dark red). See text for the definition of the drought categories.



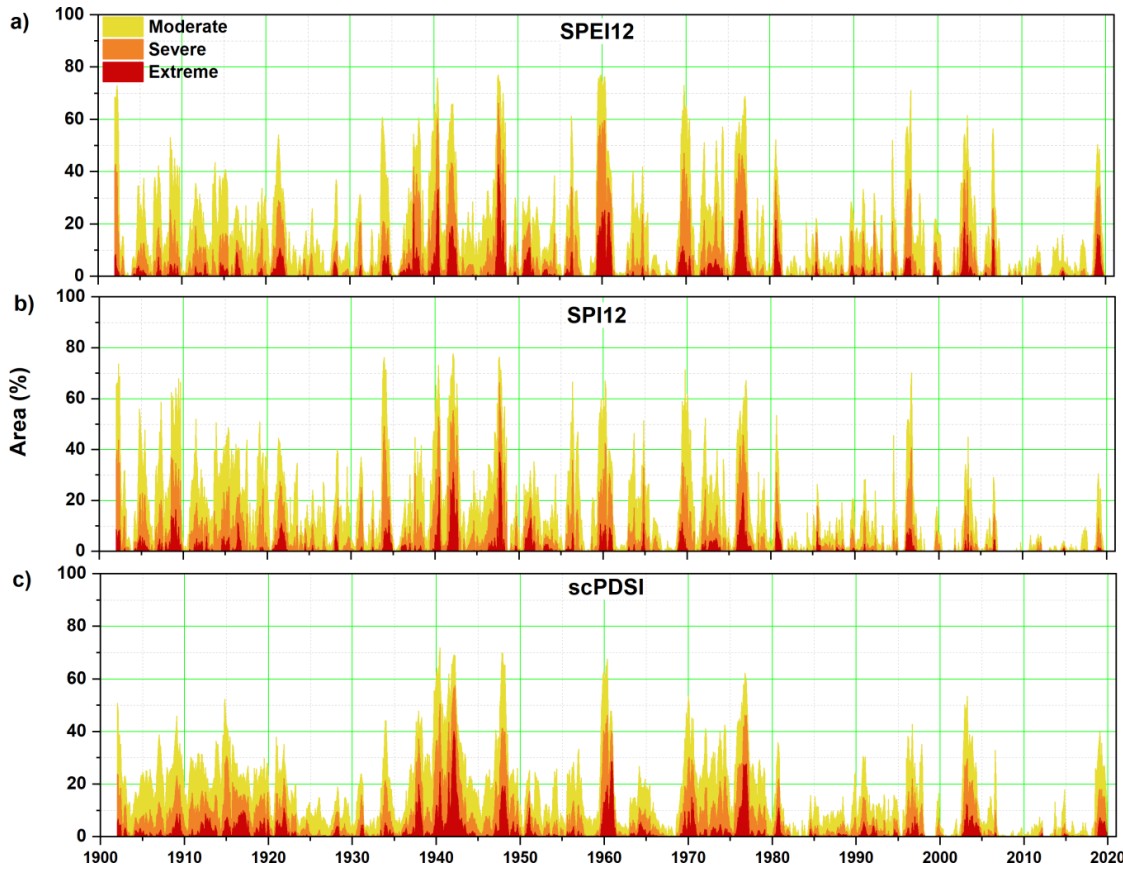

**Figure 4.** Temporal evolution of the percentage area affected by droughts over **NEU** for: a) SPEI12, b) SPI12 and c) scPDSI, for three drought severity categories: moderate (yellow), severe (orange) and extreme (dark red). See text for the definition of the drought categories.



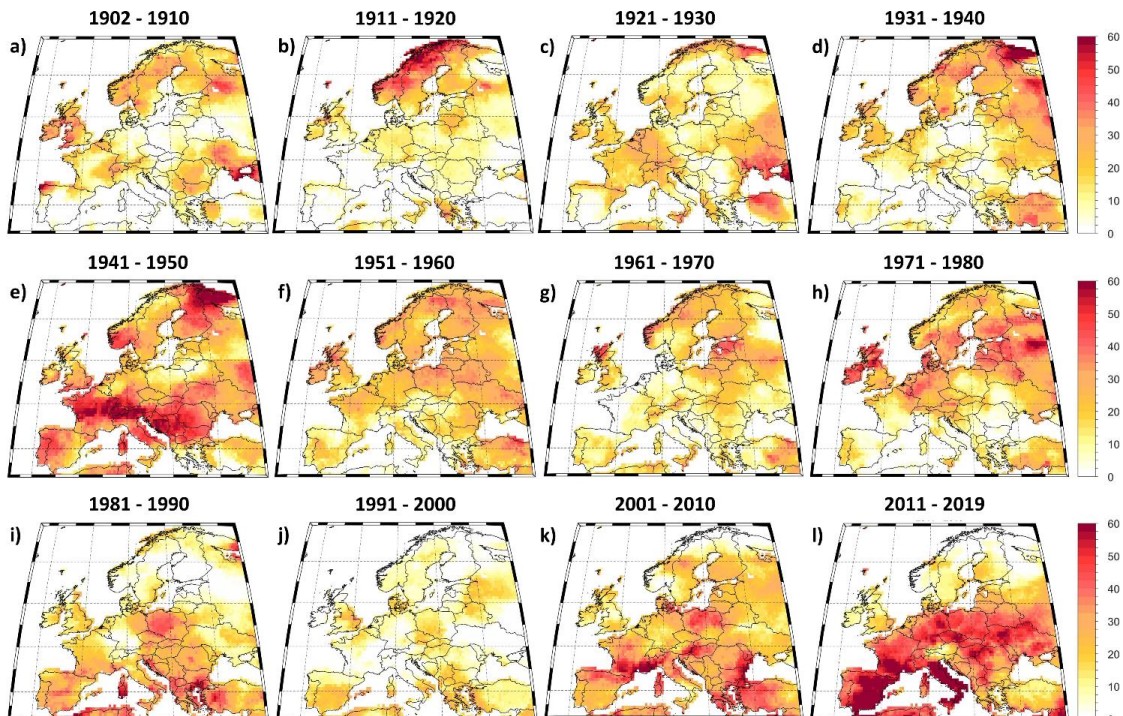

***Figure 5***. Decadal frequency of drought duration for **moderate drought** (SPEI12 between -1.0 and -1.5): a) 1902 – 1901; b) 1911 – 1920; c) 1921 – 1930; d) 1931 – 1940; e) 1941 – 1950; f) 1951 – 1960; g) 1961 – 1970; h) 1971 – 1980; i) 1981 – 1990; j) 1991 – 2000; k) 2001 – 2010 and l) 2011 – 2019. Units: number of months/period.





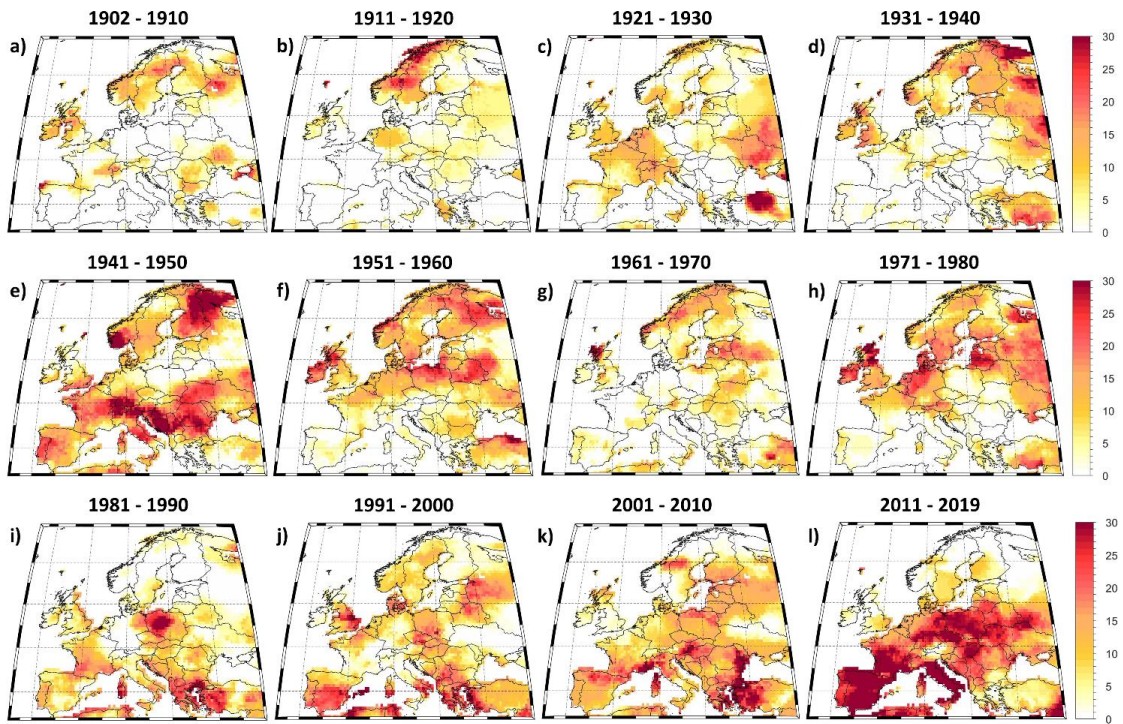

*Figure 6*. Decadal frequency of drought duration for **severe drought** (SPEI12 between -1.51 and -2): a) 1902 – 1901; b) 1911 – 1920; c) 1921 – 1930; d) 1931 – 1940; e) 1941 – 1950; f) 1951 – 1960; g) 1961 – 1970; h) 1971 – 1980; i) 1981 – 1990; j) 1991 – 2000; k) 2001 – 2010 and l) 2011 – 2019. Units: number of months/period.




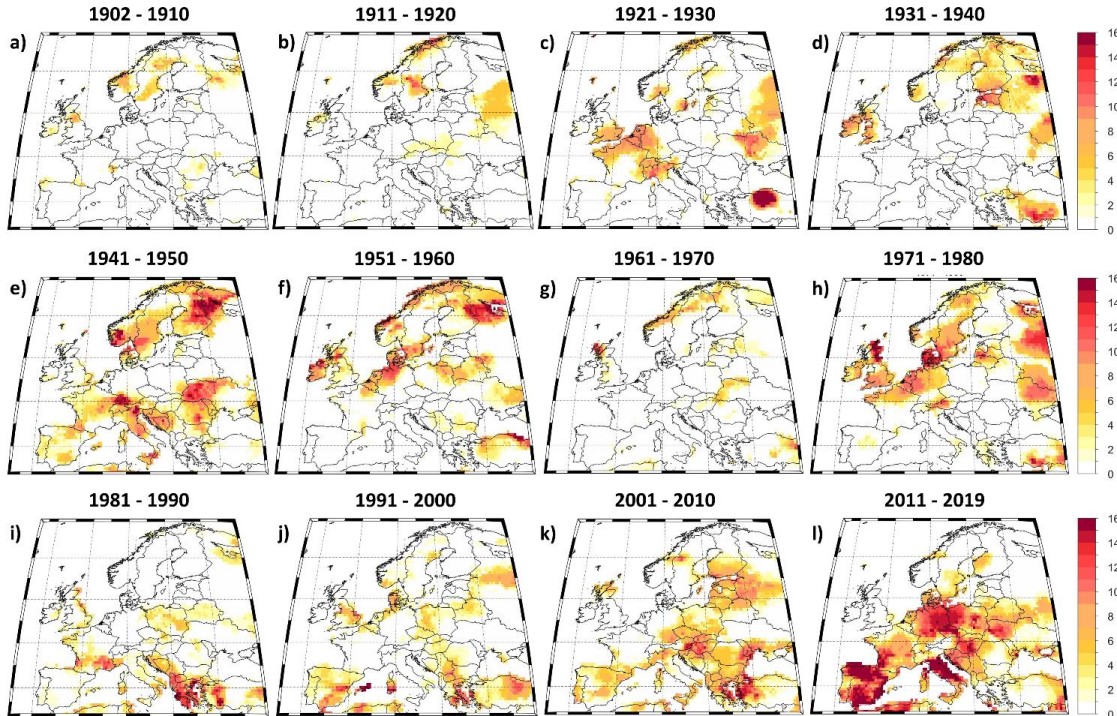

*Figure 7*. Decadal frequency of drought duration for **extreme drought** (SPEI12<-2): a) 1902 – 1901; b) 1911 – 1920; c) 1921 – 1930; d) 1931 – 1940; e) 1941 – 1950; f) 1951 – 1960; g) 1961 – 1970; h) 1971 – 1980; i) 1981 – 1990; j) 1991 – 2000; k) 2001 – 2010 and l) 2011 – 2019.  Units: number of months/period.




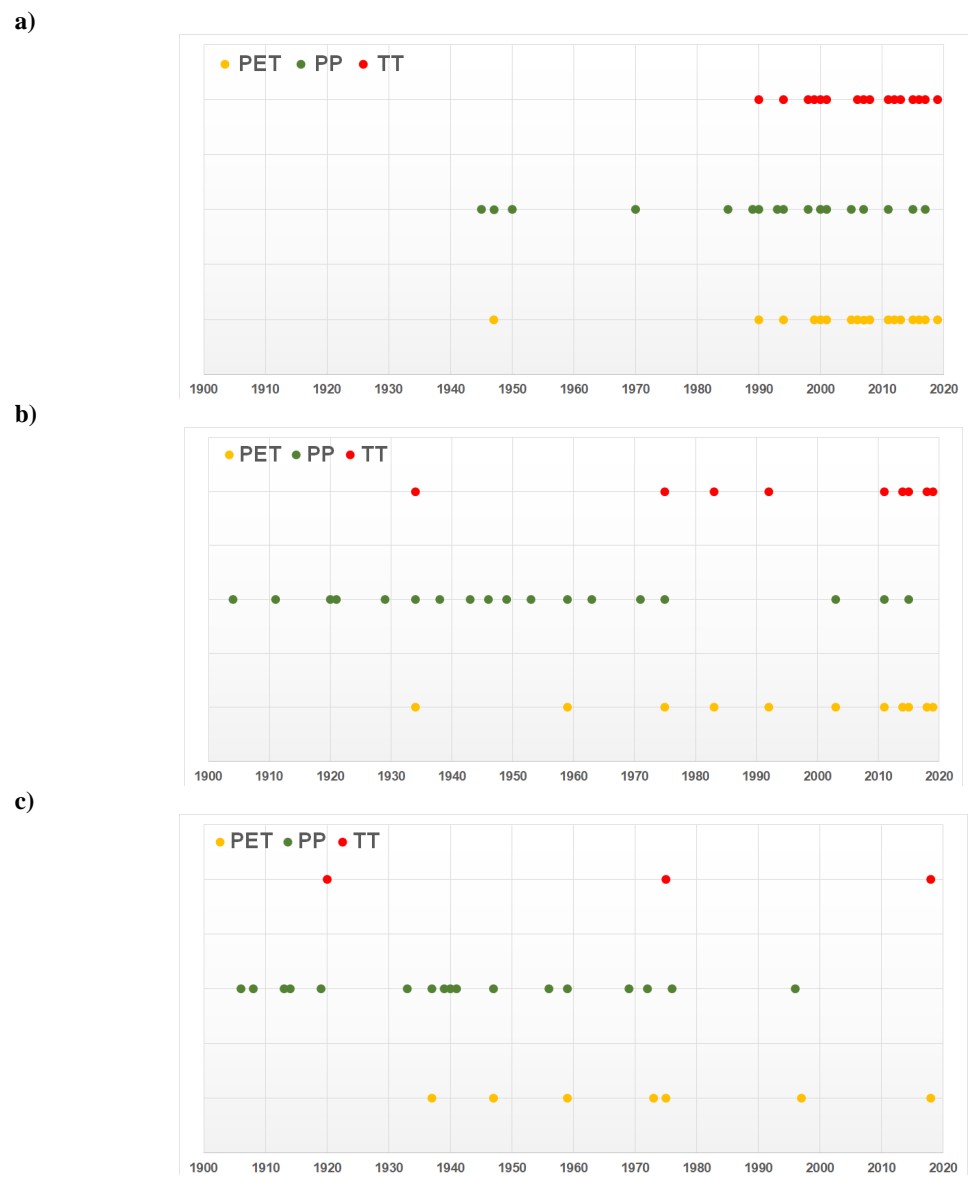


**Figure 8**. a) Occurrence of warm and dry events ($TT_{80}$/$SPEI12_{20}$ – red dots), low precipitation and dry events ($PP_{20}$/$SPEI12_{20}$ – green dots) and enhanced evaporation and dry events ($PET_{80}$/$SPEI12_{20}$ – yellow dots) for MED area; b) as in a) but for CEU and c) as in a) but for NEU. $TT_{80}$/$SPEI12_{20}$ indicates that we took into account the common years when the temperature was higher than the $80^{th}$ percentile and SPEI12 was smaller that the $20^{th}$ percentile. $PP_{20}$/$SPEI12_{20}$ indicates that we took into account the common years when the precipitation was smaller than the $20^{th}$ percentile and SPEI12 was smaller that the $20^{th}$ percentile. $PET_{80}$/$SPEI12_{20}$ indicates that we took into account the common years when the potential evapotranspiration was higher than the $80^{th}$ percentile and SPEI12 was smaller that the $20^{th}$ percentile.




*Figure 9*. The spatial extent and the year of record of the driest years, based on the monthly SPEI12, over
Europe. Analyzed period: 1902–2019.




***Figure 10***. Yearly anomalies for: a) PP - 1921, b) PP – 2019; c) TT – 1921; d) TT – 2019; e) PET– 1921 and f) PET – 2019. The anomalies are computed relative to the period 1971–2000.


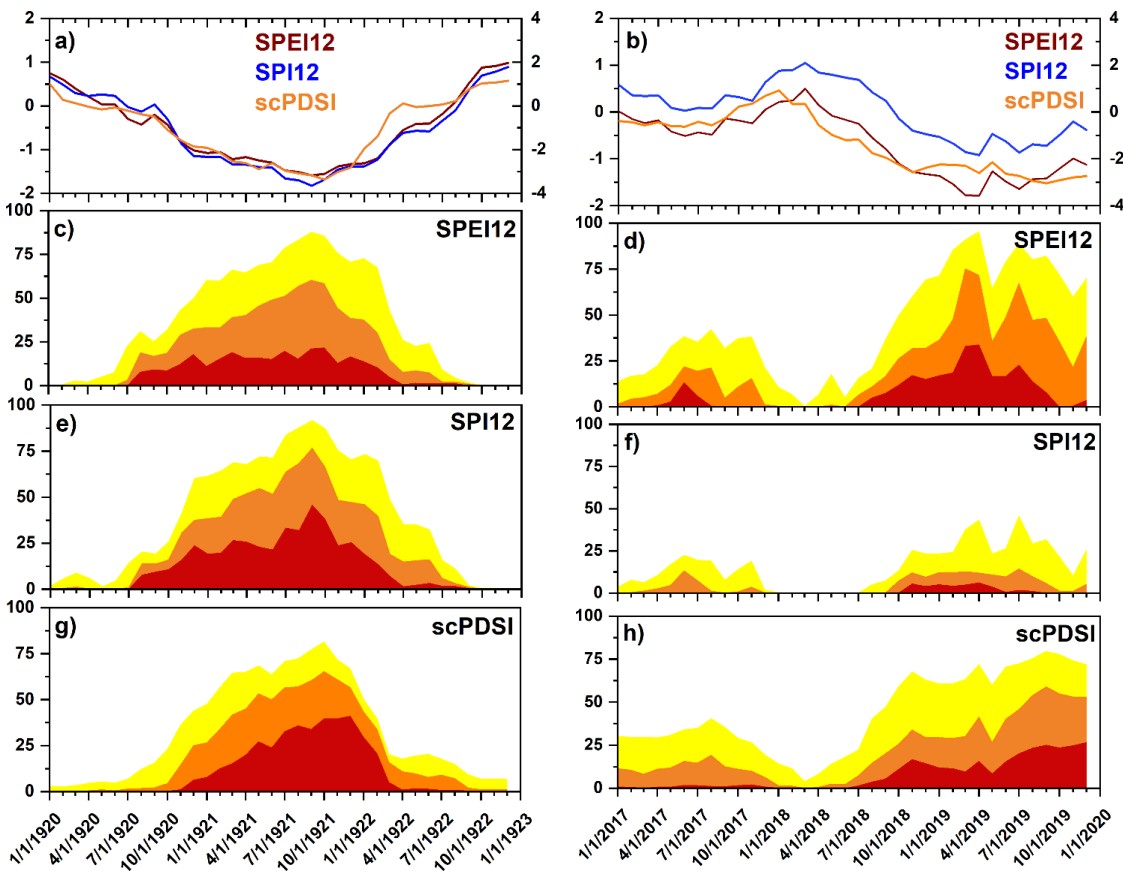

**Figure 11.** a) Temporal evolution of the monthly SPEI12 (red line); SPI12 (blue line) and scPDSI (orange line) for the period January 1920 – December 1922**;** b) as in a) but for the period January 2017 – December 2019**;** c) Temporal evolution of the drought area for SPEI12 for the period January 1920 – December 1922, for different types of dorught: moderate (yellow), severe (orange) and extreme (red); d) as in b) but for the period January 2017 – December 2019; e) Temporal evolution of the drought area for SPI12 for the period January 1920 – December 1922, for different types of dorught: moderate (yellow), severe (orange) and extreme (red); f) as in e) but for the period January 2017 – December 2019; g) Temporal evolution of the drought area for scPDSI for the period January 1920 – December 1922, for different types of dorught: moderate (yellow), severe (orange) and extreme (red) and h) as in g) but for the period January 2017 – December 2019.
