# Peer review of "Changes in drought features at European level over the last 120 years"

_Natural Hazards and Earth System Sciences, 2021_

## Author Comment (AC1)

We thank the reviewer for the constructive and detailed comments. All of them were used to improve the manuscript. Specific answers to comments are included below.

The manuscript focuses on analyzing drought in Europe in the period 1902-2019 by means of the CRU TS v4.04 dataset. The paper is very interesting and presents a good analysis, however, in my opinion there are a few drawbacks in the paper, which can be eliminated by carrying out some minor revisions following the list of comments below.

My main concern refers to the use of the CRU TS v4.04 dataset for the period 1902-2019. The numbers and locations of stations contributing to any grid cell of the dataset changed over time, especially in the first half of the past century. Can the authors provide a map showing the evolution of the stations' density in the study area? Can the authors provide a comment on how station distribution could influence the analyses shown on the maps?

We agree with this concern. In the revised version of the manuscript we will add some paragraphs regarding the distribution of the stations in the CRU TS 4.04 dataset over Europe. Unfortunately we do not have access to the distribution of the stations to make our own figure, but we can definitely refer to the Harris et al. (2020) paper for an overview of the station distribution for precipitation (Figure 1 in their paper) and temperature (Figure S1 in their paper). As clearly shown also in their paper the stations distribution over Europe is relatively homogenous even at the beginning of the 20th century, thus we believe that our results are robust throughout the analyzed period.

In the trend analysis the authors identified significant changes but they must specify the significance level considered.

We will add this information in the revised version of the manuscript.

Line 301: Figures 6-8 should be Figure 8

We will modify the text accordingly.

Finally, in the conclusions the authors added a discussion to underline the added value of their work compared to other similar in the same area, but some important comparison with drought analyses performed with gridded databases are missing. For example, in my knowledge, gridded data sets have been used for drought analyses in Europe producing maps of the self-calibrating Palmer Drought Severity Index (van der Schrier et al. 2006 doi: 10.1175/JCLI3734.1) or maps of the SPI trend at different timescale (Caloiero et al. 2018 doi: 10.3390/w10081043).

We will add the aforementioned references in the revised version of the manuscript and we will integrated the comparison with them throughout the text.

---

## Author Comment (AC2)

We thank the reviewer for the constructive and detailed comments. All of them will be used to improve the manuscript. Specific answers to comments are included below.

The aim of the manuscript is to analyze drought evolution in space and time over the period 1901-2019 in three European macro regions, namely South Europe/Mediterranean region (MED), Central Europe (CEU) and North Europe (NEU). In particular, a comparison between three different drought indices, that is SPI, SPEI and ScPDSI, is carried out.

General comments

The topic is interesting and fits with the journal aims and scopes. It is well written and organized. Although the study is not novel from a methodological viewpoint, overall, it is clear and well detailed. The results seem accurate and highlight some relevant differences in drought detection over Europe between SPI and SPEI, with special reference to those events occurred during the last decades, due to increasing temperature, and therefore evapotranspiration.

I suggest a few revisions before publication. Specific comments:

In the SPI and SPEI computation based on CRU datasets, it was assumed that monthly accumulated precipitation series were gamma distributed and the accumulated differences between monthly precipitation and potential evapotranspiration were log-logistic distributed. Although these are the probability distributions commonly used to calculate these drought indices, it would not be surprising if they did not fit all the data. Have you checked the goodness of fit of these distributions for all the grid cells series?

We have actually tried all the available distributions in the SPEI package and compared the results between all the distributions (e.g. log-Logistic, Gamma and Pearson III), but no significant changes have been noticed. Thus we have decide to show, in our manuscript, the results based on the widely used candidate distributions: Gamma for SPI and log-logistic for SPEI, respectively.

In the abstract (L 21), Section 2 (LL 150-151) and Conclusions (L 411), the authors talk about the application of a joint distribution to analyze compound events (i.e., drought and high temperature concurrent events). However, as they clarify at LL 155-158, they just calculate the number of occurrences of compound events based on fixed thresholds. This is rather different than applying a joint probability distribution (i.e., fit a bivariate or multivariate distribution) to model compound events. Therefore, I suggest the authors to replace "joint distribution" just with "frequency analysis of compound events".

We apologize for the misunderstanding. We will correct the term in the revised version of the manuscript.

In the compound analysis of droughts and high temperature, the occurrence of these events is based on fixed thresholds (e.g., $80^{th}$ percentile for temperature and $20^{th}$ for SPEI). How these thresholds have been chosen? Besides, I wonder if a sensitivity analysis has been carried out by changing the values of these thresholds. Please provide details.

We agree with the reviewer's comment and in the revised version of the manuscript we will add some information regarding the use of other thresholds. We have tested threshold of $70^{th}$, $75^{th}$, $80^{th}$ and $85^{th}$ percentile for temperature and $15^{th}$, $20^{th}$, $25^{th}$ and $30^{th}$ percentile for SPEI, but we could not find any significant change in the compound analysis. We chose the $20^{th}$ (SPEI) and $80^{th}$ (TT) percentile to have enough extreme events to analyze.

There are some previous studies on drought analysis at European level identifying similar trends, which are not cited in the manuscript. For the sake of completeness, the authors should include for instance:

Oikonomou, P.D., Karavitis, C.A., Tsesmelis, D.E., Kolokytha, E., Maia, R. Drought Characteristics Assessment in Europe over the Past 50 Years (2020). DOI: 10.1007/s11269-020-02688-0
Hänsel, S., Ustrnul, Z., Łupikasza, E., Skalak, P. Assessing seasonal drought variations and trends over Central Europe (2019). DOI: 10.1016/j.advwatres.2019.03.005
Christoph C. Raible, Oliver Bärenbold & Juan José Gómez-navarro (2017) Drought indices revisited – improving and testing of drought indices in a simulation of the last two millennia for Europe, Tellus A: Dynamic Meteorology and Oceanography, 69:1, 1287492, DOI:10.1080/16000870.2017.1296226
Bonaccorso, B., Peres, D.J., Cancelliere, A., Rossi, G. Large Scale Probabilistic Drought Characterization Over Europe (2013). DOI: 10.1007/s11269-012-0177-z
Parry, S., Hannaford, J., Lloyd-Hughes, B., Prudhomme, C. Multi-year droughts in Europe: Analysis of development and causes (2012). DOI: 10.2166/nh.2012.024
Bordi, I., Fraedrich, K., Sutera, A. Observed drought and wetness trends in Europe: An update (2009). DOI: 10.5194/hess-13-1519-2009
With reference to the potential interconnection between droughts and heatwaves, I believe that the discussion could benefit by the comparison with the results of the following studies:
Markonis, Y., Kumar, R., Hanel, M., Rakovec, O., Máca, P., Kouchak, A.A. The rise of compound warm-season droughts in Europe (2021) . DOI: 10.1126/sciadv.abb9668
Bezak, N., Mikoš, M. Changes in the compound drought and extreme heat occurrence in the 1961–2018 period at the european scale (2020). DOI: 10.3390/w12123543
Samaniego, L., Thober, S., Kumar, R. et al. Anthropogenic warming exacerbates European soil moisture droughts. Nature Clim Change 8, 421–426 (2018). https://doi.org/10.1038/s41558-018-0138-5
In the revised version of the manuscript, we will try to add the aforementioned references and integrated them throughout the text.

Minor comments
LL 91-92: provide references at the end of the sentence.
We will add some references at the end of the sentence.

L 234: extend must be extent.
The text will be modified as suggested.

L 282: add "are" before relatively.
The text will be modified as suggested.

L 294 and L 420: add "of" after consideration.
The text will be modified as suggested.

L 329: To most must be The most.
The text will be modified as suggested.

L 361: Change SEPI in SPEI.
The text will be modified as suggested.

L 435: projected.
The text will be modified as suggested.

---

## Editor Decision (ED1)

**Supplementary file**

**Changes in drought features at European level over the last 120 years**

Monica Ionita[1*] and Viorica Nagavciuc[1,2]

[1] Alfred Wegner Institute Helmholtz Center for Polar and Marine Research, Bremerhaven, Germany

[2] Forest Biometrics Laboratory – Faculty of Forestry, "Stefan cel Mare" University of Suceava, Suceava, Romania

[*] Corresponding author: Monica Ionita ([Monica.Ionita@awi.de](mailto:Monica.Ionita@awi.de))

To analyzed if there are significant changes in the SPEI12, SPI12, scPDSI (Figure1 and Figure S2), PP, TT and PET (Figure S3) and the drought area (Figures 3 - 5) we have used the rank-based non-parametric Mann-Kendall (M-K) test and Spearman's Rho (Mann, 1945; Kendall, 1948), which are less sensitive to outliers than parametric statistics, were used. To avoid the influence of serial persistence on M-K test results, the modified M-K (MMK) trend test was used, using the computation algorithm discussed by Hamed and Rao (1998).

*Table S1.* Linear trends of the drought area for different drought types (moderate, severe and extreme) for SPEI12, SPI12 and scPDSI for the three analyzed regions: MED, CEU and NEU.

| | SPEI12 | | | SPI12 | | | scPDSI | | |
|---|---|---|---|---|---|---|---|---|---|
| | Moderate[i] | Severe[ii] | Extreme[iii] | Moderate[i] | Severe[ii] | Extreme[iii] | Moderate[i] | Severe[ii] | Extreme[iii] |
| **MED** | ↑* | ↑* | ↑* | ↑ | ↑ | ↑ | ↑* | ↑* | ↑* |
| **CEU** | ↑* | ↑* | ↑* | ↓ | ↓ | ↓ | ↑* | ↑* | ↑* |
| **NEU** | ↓* | ↓* | ↓ | ↓* | ↓* | ↓* | ↓* | ↓* | ↓* |

↑* - indicates a significant positive trend (99% significance level);
↑  - indicates a positive, but not significant trend;
↓*- indicates a significant negative trend (99% significance level);
↓  - indicates a negative, but not significant trend;

i) moderate drought (SPI/SPEI values between -1 and -1.5 and scPDSI values between -2 and -3);
ii) severe drought (SPI/SPEI values between -1.5 and -2 and scPDSI values between -3 and -4);
iii) extreme drought (SPI/SPEI values less than -2 and scPDSI values smaller than -4).

[Figure]

***Figure S1.*** Spatial delimitation of the macro regions analyzed in this study: South Europe/ Mediterranean region (MED); Central Europe (CEU) and North Europe (NEU). Data source for the digital elevation model: (NOAA, 2009).

[Figure]

***Figure S2.*** a) Linear trend of February SPEI3; b) as in a) but for SPI3; c) linear trend of May SPEI3; d) as in c) but for SPI3; e) linear trend of August SPEI3; f) as in e) but for SPI3; g) linear trend of SPEI3 November and h) as in g) but for SPI3. Stipples indicate statistically significant trends. Analyzed period 1902 – 2019. Units: z-scores/ 118 years.

[Figure]

***Figure S3.*** a) Linear trend of winter (DJF) potential evapotranspiration (PET); b) as in a) but for the winter (DJF) precipitation (PP); c) as in a) but for the winter (DJF) mean air temperature (TT); d) as in a) but for spring (MAM); e) as in b) but for spring (MAM); f) as in c) but for spring (MAM); g) as in a) but for summer (JJA); h) as in b) but for summer (JJA); i) as in c) but for summer (JJA); j) as in a) but for autumn (SON); k) as in b) but for autumn (SON) and l) as in c) but for autumn (SON). Stipples indicate statistically significant trends. Analyzed period 1902 – 2019. Units: PET (mm/decade), PP (mm/decade) and TT (°C/decade).

[Figure]

***Figure S4.*** Decadal frequency of drought duration for **moderate drought** (SPI12 between -1.0 and -1.5): a) 1902 – 1901; b) 1911 – 1920; c) 1921 – 1930; d) 1931 – 1940; e) 1941 – 1950; f) 1951 – 1960; g) 1961 – 1970; h) 1971 – 1980; i) 1981 – 1990; j) 1991 – 2000; k) 2001 – 2010 and l) 2011 – 2019. Units: number of months/period.

[Figure]

***Figure S5.*** Decadal frequency of drought duration for **severe drought** (SPI12 between -1.51 and -2): a) 1902 – 1901; b) 1911 – 1920; c) 1921 – 1930; d) 1931 – 1940; e) 1941 – 1950; f) 1951 – 1960; g) 1961 – 1970; h) 1971 – 1980; i) 1981 – 1990; j) 1991 – 2000; k) 2001 – 2010 and l) 2011 – 2019. Units: number of months/period.

[Figure]

***Figure S6.*** Decadal frequency of drought duration for **extreme drought** (SPI12<-2): a) 1902 – 1901; b) 1911 – 1920; c) 1921 – 1930; d) 1931 – 1940; e) 1941 – 1950; f) 1951 – 1960; g) 1961 – 1970; h) 1971 – 1980; i) 1981 – 1990; j) 1991 – 2000; k) 2001 – 2010 and l) 2011 – 2019.  Units: number of months/period.

[Figure]

***Figure S7.*** a) Time series of the annual precipitation (PP) averaged over MED; b) as in a) but for the potential evapotranspiration (PET); c) as in a) but for mean air temperature (TT) and d) as in a) but for SPEI12. The red line in a) – d) indicates the 21 years running mean.

[Figure]

***Figure S8.*** a) Time series of the annual precipitation (PP) averaged over CEU; b) as in a) but for the potential evapotranspiration (PET); c) as in a) but for mean air temperature (TT) and d) as in a) but for SPEI12. The red line in a) – d) indicates the 21 years running mean.

[Figure]

***Figure S9.*** a) Time series of the annual precipitation (PP) averaged over MED; b) as in a) but for the potential evapotranspiration (PET); c) as in a) but for mean air temperature (TT) and d) as in a) but for SPEI12. The red line in a) – d) indicates the 21 years running mean.

[Figure]

***Figure S10***. a) Occurrence of warm and dry events ($TT_{75}$/$SPEI12_{25}$ – red dots), low precipitation and dry events ($PP_{25}$/$SPEI12_{25}$ – green dots) and enhanced evaporation and dry events ($PET_{75}$/$SPEI12_{25}$ – yellow dots) for MED area; b) as in a) but for CEU and c) as in a) but for NEU. $TT_{75}$/$SPEI12_{25}$ indicates that we took into account the common years when the temperature was higher than the 75th percentile and SPEI12 was smaller that the 25th percentile. $PP_{25}$/$SPEI12_{25}$ indicates that we took into account the common years when the precipitation was smaller than the 25th percentile and SPEI12 was smaller that the 25th percentile. $PET_{75}$/$SPEI12_{25}$ indicates that we took into account the common years when the potential evapotranspiration was higher than the 75th percentile and SPEI12 was smaller that the 25th percentile.

[Figure]

**Figure 11.** a) Occurrence of warm and dry events ($TT_{90}/SPEI12_{10}$ – red dots), low precipitation and dry events ($PP_{10}/SPEI12_{10}$ – green dots) and enhanced evaporation and dry events ($PET_{90}/SPEI12_{10}$ – yellow dots) for MED area; b) as in a) but for CEU and c) as in a) but for NEU. $TT_{90}/SPEI12_{10}$ indicates that we took into account the common years when the temperature was higher than the 90th percentile and SPEI12 was smaller that the 10th percentile. $PP_{10}/SPEI12_{10}$ indicates that we took into account the common years when the precipitation was smaller than the 10th percentile and SPEI12 was smaller that the 10th percentile. $PET_{90}/SPEI12_{10}$ indicates that we took into account the common years when the potential evapotranspiration was higher than the 90th percentile and SPEI12 was smaller that the 10th percentile.

[Figure]

***Figure S12.*** Spatial evolution of the SPEI12 between November 1920 until January 1922.

[Figure]

***Figure S13.*** The spatial extent and the year of record of the driest years, based on the monthly SPI12, over Europe. Analyzed period: 1902–2019.

[Figure]

***Figure S14.*** Spatial evolution of the SPEI12 between October 2018 until December 2019.

**References**

Hamed, K. H. & Rao, A. R. A modified Mann-Kendall trend test for autocorrelated data. *Journal of Hydrology* **204**, 182–196 (1998).

Kendall, M.G., 1948. Rank Correlation Methods. Griffin, London

Mann, H. B., 1945. Nonparametric tests against trend. *Econometrica* **13**, 245–259.

NOAA: ETOPO1 1 Arc-Minute Global Relief Model, Cent. NOAA Natl. Geophys. Data [online] Available from: https://www.ngdc.noaa.gov/mgg/global/ (Accessed 18 October 2020), 2009

---

## Author Response (AR2)

Dear Dr. Peres,

Thank you for the suggestions and for the review process of our manuscript.

Regrading the suggested detailed analysis we have some points:

1. Probability distribution goodness of fit tests.

We did computed the SPI and SPEI by applying different distributions, but we did not wanted to make this discussion in the manuscript, mainly because such a discussion usually is the topic of a whole manuscript. Since we did not find any significant changes when using different distributions, we proceeded with the manuscript by using the well-know distributions: Gamma for SPI and log-logistic for SPEI. The correlation coefficients between the SPI/SPEI indices based on different distributions varies between 0.98 and 0.99, thus we are 100% that our choice is suitable for the current study. We added to correlation coefficients in the revised version of the manuscript, but we have decided not to add any additional figures, mainly because the paper is already very long and we have already 25 figures (main + supplementary).

2. Threshold for defining events.

To comply with the requirements from the reviewers and the editor we have added in the supplementary file 2 extra figures (Figure S10 and S11) in which we perform the same analysis as in Figure 8, but by using different thresholds for the computation of the joint occurrence of warm and dry events.

We hope that we were able to comply with the suggestions from the editor and that our manuscript is now suitable for publications.

Best wishes,

Monica Ionita